# Bicine promotes rapid formation of β-sheet-rich amyloid-β fibrils

Hye Yun Kim[1,2], HeeYang Lee[1,2,3], Jong Kook Lee[4], Hyunjin Vincent Kim[4], Key-Sun Kim[4], YoungSoo Kim [1,2,3]*

1 Department of Pharmacy, Yonsei University, Incheon, Republic of Korea, 2 Yonsei Institute of Pharmaceutical Sciences, Yonsei University, Incheon, Republic of Korea, 3 Integrated Science and Engineering Division, Yonsei University, Incheon, Republic of Korea, 4 Brain Science Institute, Korea Institute of Science and Technology, Seoul, Republic of Korea

* y.kim@yonsei.ac.kr

## Abstract

Fibrillar aggregates of amyloid-β (Aβ) are the main component of plaques lining the cerebrovasculature in cerebral amyloid angiopathy. As the predominant Aβ isoform in vascular deposits, $A\beta_{40}$ is a valuable target in cerebral amyloid angiopathy research. However, the slow process of $A\beta_{40}$ aggregation *in vitro* is a bottleneck in the search for Aβ-targeting molecules. In this study, we sought a method to accelerate the aggregation of $A\beta_{40}$ *in vitro*, to improve experimental screening procedures. We evaluated the aggregating ability of bicine, a biological buffer, using various in vitro methods. Our data suggest that bicine promotes the aggregation of $A\beta_{40}$ with high speed and reproducibility, yielding a mixture of aggregates with significant β-sheet-rich fibril formation and toxicity.

## Introduction

In the brain of patients afflicted with cerebral amyloid angiopathy (CAA), depositions of amyloid-β (Aβ) are found in the walls of the cortical and leptomeningeal vasculature [1,2]. While Aβ monomers of various lengths are produced by sequential β- and γ-secretase processing of membrane-bound amyloid-β protein precursors [3,4], $A\beta_{40}$ is the predominant isoform in vascular deposits [5]. As increasing vascular accumulations of $A\beta_{40}$ has shown to reflect the severity of CAA [6], $A\beta_{40}$ is a valuable target in CAA diagnosis and disease-modifying therapy. However, due to the lack of the hydrophobic Ile-Ala residues that facilitates the relatively fast folding of $A\beta_{42}$ [7], the process of obtaining $A\beta_{40}$ aggregates *in vitro* is time-consuming, hindering the screening process of $A\beta_{40}$-targeting molecules [8,9]. Thus, biochemical approaches to promptly and reliably access $A\beta_{40}$ aggregates can improve the experimental and clinical investigation of CAA.

Physio-chemical environmental factors, including the pH, temperature, ionic strength, and the presence of cosolvents, influence the nonpolar interactions involved in the intermolecular β-sheet assembly characteristic of amyloid fibrils [10]. Buffer ions are known to mediate amyloid fibril morphologies through discrete and specific interactions with amino acid side chains of Aβ, inducing distinct patterns of inter- and intra-residue interactions [11,12]. Optimization of *in vitro* incubation conditions can lead to higher yields of Aβ aggregates. In our previous

**Data Availability Statement:** All relevant data are within the paper and its Supporting Information files.

**Funding:** This work was supported by National Research Foundation of Korea (NRF-

2018R1A6A1A03023718 and NRF-2018R1D1A1B07048857), and POSCO Science Fellowship of POSCO TJ Park Foundation. The funders had no role in study design, data collection and analysis, decision to publish, or preparation of the manuscript.

**Competing interests:** The authors have declared that no competing interests exist.

study, we observed the differing regulatory effects of organic amino acid compounds and their derivatives on the *in vitro* fibrillation process of $A\beta_{40}$ [8]. Bicine (*N,N*-Bis(2-hydroxyethyl)glycine) is a zwitterionic buffer from Good's list, commonly used in biological and biochemical experiments [13,14]. Here, we report that bicine promotes the formation of β-sheet-rich $A\beta_{40}$ fibrils *in vitro*. We investigated the accelerating effect of bicine on the propensity of $A\beta_{40}$ aggregation in comparison to phosphate buffered saline (PBS), by using thioflavin T (ThT) fluorescence assays, far-ultraviolet circular dichroism (CD) spectroscopy, and transmission electron microscopy (TEM). Time-dependent changes in the size composition of $A\beta_{40}$ species upon incubation with bicine was monitored via SDS-PAGE followed by photo-induced cross-linking of unmodified proteins (PICUP) and silver staining. Cell viability assays were conducted to assess the toxicity of the aggregated peptides.

## Materials and methods

### Materials

$A\beta_{40}$, $H_2N$-DAEFRHDSGYEVHHQKLVFFAEDVGSNKGAIIGLMVGGVV-COOH, was synthesized and purified using a stepwise solid phase peptide synthesis protocol as previously reported [15]. Bicine, PBS, 1,1,1,3,3,3-hexafluoro-2-propanesulfonic acid (HFIP), anhydrous dimethylsulfoxide (DMSO), Tris(2,2'bipyridyl)dichlororuthenium(II) (Ru(Bpy)) and ammonium persulfate (APS) were obtained from Sigma-Aldrich. SDS-PAGE apparatus, 15% SDS-PAGE gel, and 10X Tris-glycine SDS Buffer were purchased from Bio-Rad. PBS (1X) consists of phosphate buffer concentration of 0.01M and a sodium chloride concentration of 0.154M. Lane marker reducing sample buffer (5X) was from Thermo Scientific and protein marker was purchased GE Healthcare. Silver staining kit was acquired from Amersham Biosciences. All other reagents, solvents, and chemicals were of the highest purity commercially available and used as received.

### Preparation and aggregation of $A\beta_{40}$

Lyophilized $A\beta_{40}$ peptides were dissolved in HFIP to obtain homogenous monomeric peptides [16]. The amyloid peptide solution (0.5 mM) in pre-chilled HFIP was sonicated and vortexed gently for 5 and 30 minutes, respectively. After the complete removal of HFIP, the peptides were dissolved in DMSO to make a stock solution of Aβ (5 mM). Aggregation of $A\beta_{40}$ (50 μM final concentration) was initiated by diluting and incubating the stock in bicine or PBS at 37°C at different periods: 2, 4, and 7 days. The 0 day $A\beta_{40}$ sample was dissolved in PBS and immediately stored at −70°C.

### ThT assay

To examine the propensity of $A\beta_{40}$ aggregation in bicine, we performed concentration- and time-dependent ThT assays. For the concentration-dependent assay, $A\beta_{40}$ (50 μM) was incubated in serial concentrations of bicine (78.130, 156.25, 312.50, 625.00, 1250.0, 2500.0, 5000.0, 10000, 20000 μM). For the time-dependent assay, $A\beta_{40}$ (50 μM) was incubated in bicine or PBS at different periods: 2, 4, and 7 days. Each incubated sample (25 μL) was mixed with ThT solution (75 μL, 5 μM in pH 8.5 glycine buffer) in a half-area black 96-well plate [17]. The pH of the incubated mixtures was adjusted to pH 5.5 before the addition of ThT solution due to the pH dependent binding affinity between ThT and Aβ [18]. Shifted fluorescence was measured on an Envision 2103 multilabel reader (Perkin-Elmer) at 450 nm (ex) and 480 nm (em).

## CD

Far UV CD spectra diluted samples of 25 μM $A\beta_{40}$ incubated in bicine for 2 days and in PBS for 0 and 2 days were recorded with a Jasco J-715 Spectropolarimeter using 1 mm path-length cells (Hellma). Measurements were carried out from 260 to 190 nm at 37˚C with a data pitch of 0.1 nm, bandwidth of 2.0 nm, scan sped of 20 nm/minute and response time of 8 seconds.

## TEM

$A\beta_{40}$ (50 μM) incubated for 2 days in PBS and bicine were negatively stained on a cleaned grid and subsequently visualized via TEM. Samples were prepared as previously described [19]. Imaging analysis was performed using a Philips CM-30 transmission electron microscope.

## SDS-PAGE with PICUP

To analyze the size distribution of the incubated amyloid samples in detail, PICUP chemistry followed by SDS-PAGE was employed [16]. APS (0.7 μL, 20 mM) and Ru(Bpy) (0.7 μL, 1 mM) in a sodium phosphate buffer (10 mM, pH 7.4) were added to $A\beta_{40}$ (50 μM) samples incubated in 12 μL of bicine or PBS. For the cross-linking of peptides, mixtures were irradiated, 3 times (1 second each), using a 200-watt incandescent lamp and quenched with a reducing sample buffer. Cross-linked $A\beta_{40}$ samples were loaded onto a 1.0 mm-thick 10–20% tris-tricine gradient gel. Electrophoresis was performed at 120 volts for 1 hour and 45 minutes using a tris/tricine/SDS running buffer. After the separation step, Aβ containing gels were stained via the silver staining kit to visualize peptide bands. Each gel was developed for 10 minutes prior to the addition of a stop solution.

## Cell culture and MTT cell viability assay

The toxicity of $A\beta_{40}$ incubated in bicine was assessed using the colorimetric MTT cell viability assay as previously reported [8]. The mouse hippocampal cell line (HT-22) [20] was purchased from the Korean Cell Line Bank (Seoul National University, Republic of Korea). Cells were cultured in DMEM supplemented with 10% fetal bovine serum and 1% penicillin-streptomycin at 37˚C in a 5% $CO_2$ incubator. HT-22 cells were seeded into 96-well plates ($3\times10^3$ cells/ well). $A\beta_{40}$ (50 μM) incubated in bicine (2, 4, and 7 days) or PBS (0, 2, 4 and 7 days) were diluted 50 folds with cell starvation medium and then treated to HT-22 cells. Non-treated cells with starvation media were used as a control. After 20 hours, the MTT assay was performed. Absorption values at 570 nm were measured using an Envision 2103 multilabel reader (Perkin-Elmer).

## Results

The concentration- and time-dependent effects of bicine on the fibril formation of $A\beta_{40}$ peptides were first examined through ThT fluorescence assays. Direct binding of ThT to amyloid β-sheet structures induces a characteristic change in fluorescence intensity [9]. To analyze the propensity of amyloid fibril formation in increasing buffer concentrations, we incubated monomeric $A\beta_{40}$ peptides (50 μM) in serial concentrations of bicine (78.130 μM to 20.000 mM) at 37˚C for 2 days. Bicine was found to effectively promote the formation of β-sheet-rich fibrils in a short period of time ($EC_{50}$ = 837 nM) (Fig 1A). Fluorescence intensity increased in a concentration-dependent manner. As 20 mM was the highest concentration that did not show cytotoxicity (S1 Fig), 20 mM of bicine was used in the following experiments. Subsequently, time-dependent ThT assays were performed to examine the rate of bicine-induced fibrillation (Fig 1B). $A\beta_{40}$ monomers were incubated in either bicine (20 mM) or control PBS

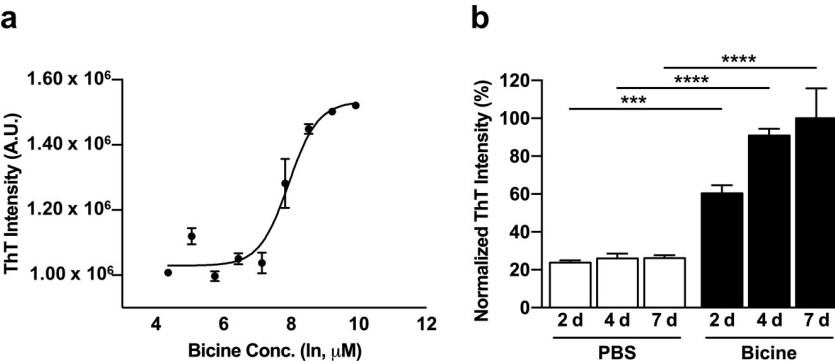

**Fig 1. Fibrillation kinetics of Aβ(1–40) aggregation in bicine solution.** ThT fluorescence assay was used to examine the propensity of fibril formation affected by bicine concentration (**a**) and incubation time (**b**). (**a**) Aβ(1–40) (50 μM) peptides were dissolved and incubated in serial concentrations of bicine (78.130, 156.25, 312.50, 625.00, 1250.0, 2500.0, 5000.0, 10000, 20000 μM, presenting ln concentration) for 2 days at 37˚C. (**b**) Aβ(1–40) (50 μM) peptides were dissolved and incubated in bicine (20 mM) or PBS (1X) at 3 time points: 2, 4 and 7 days. ThT fluorescence intensity was normalized to the Aβ(1–40) sample incubated in bicine for 7 days (100%). These experiments were performed in triplicate independently. Error bars indicate the standard deviation. Statistical analysis was performed using one-way ANOVA followed by Tukey's multiple comparison test. ***$P \leq 0.001$, ****$P \leq 0.0001$. ThT: thioflavin T, A.U.: arbitrary unit, Conc.: concentration, d: day.

solutions at 37˚C for 2, 4 and 7 days and ThT fluorescence intensities were scanned. We found that 2 days of incubation in bicine were enough to effectively form significantly higher levels of Aβ$_{40}$ fibrils than those incubated in PBS of the same duration. Bicine was shown to continuously induce fibrillation for 7 days while no substantial increase of fibril levels was observed in the PBS solution.

To confirm whether 2 days of incubation in bicine gives rise to sufficient formation of mature fibrils, we performed far-ultraviolet CD and TEM. Far-ultraviolet CD was used to assess the secondary structure of bicine-induced Aβ aggregates under non-denaturing aqueous conditions. In PBS, Aβ$_{40}$ samples at 0 days and 2 days of incubation showed a residue ellipticity minimum at about 200 nm, indicating prevalent random-coil structures (Fig 2A). In contrast, in the presence of bicine (20 mM), Aβ$_{40}$ after 2 days of incubation exhibited a single negative band with minimum at 216 nm, a typical CD spectrum of the well-defined β-sheet structure [21]. Furthermore, substantial amounts of elongated amyloid fibrils were observed via TEM in

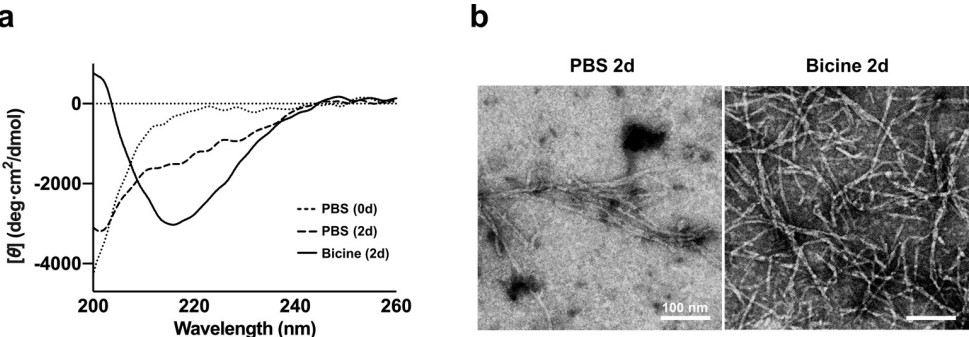

**Fig 2.** Analysis of Aβ(1–40) peptides in bicine and PBS via (a) structural analysis by CD spectra and (b) TEM imaging. (**a**) Aβ(1–40) (25 μM) peptides were dissolved and incubated in bicine (20 mM) for 2 days or PBS (1X) for 0 and 2 days. (**b**) Aβ(1–40) (50 μM) incubated for 2 days in bicine (20 mM) and PBS. TEM imagery scale bar indicates 100 nm of size. These experiments were performed in triplicate independently. CD: circular dichroism, TEM: transmission electron microscopy, d: day.

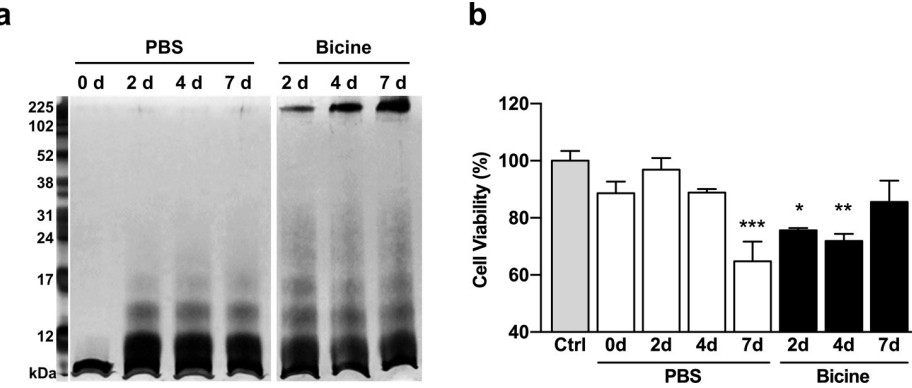

**Fig 3.** Assessing Aβ oligomeric aggregates via (a) electrophoresis and (b) MTT cell viability assay. Aβ(1–40) (50 μM) peptides were dissolved and incubated in bicine (20 mM) or PBS (1X) at 3 time points: 2, 4 and 7 days. The 0 day Aβ (1–40) sample was dissolved in PBS and immediately stored at −70˚C before use. (**a**) Aβ aggregates were visualized by silver staining after SDS-PAGE with PICUP. Aβ samples in bicine and PBS were run on separate gels; original uncropped and unadjusted images of gels are provided in Supporting Information (S1 Fig). (**b**) Aβ aggregates were treated to HT-22 cells. Non-treated cells were used as a control. These experiments were performed in triplicate independently. Error bars indicate the standard deviation. Statistical analysis was performed using one-way ANOVA followed by Tukey's multiple comparison test. $^{*}P \leq 0.05$, $^{**}P \leq 0.01$, $^{***}P \leq 0.001$. SDS-PAGE: sodium dodecyl sulfate-polyacrylamide gel electrophoresis, PICUP: photo-induced cross-linking of unmodified proteins, Ctrl: control, d: day.

Aβ$_{40}$ samples incubated in bicine, while fibril formation was sparser in PBS (Fig 2B). Overall, these results show that 2 days of incubation in bicine sufficiently promoted the formation of β-sheet-rich Aβ$_{40}$ fibrils with elongated morphology *in vitro*.

As soluble Aβ oligomers and protofibrils have been identified as the pathogenic culprits of toxicity in AD brains [22,23], we further examined the effect of bicine on the formation and toxicity of oligomeric Aβ$_{40}$. SDS-PAGE with PICUP chemistry and silver-staining was used to visualize the time-dependent changes in the size distribution of amyloid aggregates treated with bicine (Fig 3A). Monomeric Aβ$_{40}$ peptides were incubated with bicine (20 mM) or PBS for 2, 4 and 7 days. While similar amounts of Aβ$_{40}$ dimers and trimers were formed in both PBS and bicine, higher size oligomers (above 17 kDa) were notably increased in samples incubated in bicine. These oligomer bands did not change visibly throughout increasing incubation periods. However, we observed significant time-dependent development of high molecular weight aggregates (above 200 kDa) in bicine. These high molecular weight aggregates were not detected in PBS. Next, in order to assess the toxicity of Aβ$_{40}$ incubated in bicine, colorimetric MTT assay was performed using HT-22 cells (Fig 3B) [8]. Upon treatment of preincubated Aβ$_{40}$ samples to HT-22 cells, we found that bicine had significantly increased Aβ$_{40}$ cytotoxicity within 2 days. Interestingly, we found that Aβ$_{40}$ in bicine for 7 days was not significantly different than the control, indicating a decrease in amyloid toxicity. As SDS-PAGE results show that prolonged bicine-treatment induced a proportional increase of fibrils, the reduced toxicity may be due to the conversion of toxic Aβ oligomers into relatively less toxic fibrils. These results indicate that the fibril-to-oligomer composition as well as the toxicity of Aβ$_{40}$ in bicine can be controlled through modulation of the incubation period.

## Discussion

Many endeavours to regulate the Aβ misfolding *in vitro* indicate that biochemical environments highly influence the degree and morphology of protein aggregation [8,24]. Due to the versatile roles of buffer ions in amyloid fibrillogenesis [8–11], adequate buffer selection is

crucial for *in vitro* experiments. In this study, we demonstrate that bicine accelerates the formation of β-sheet-rich fibrils of $A\beta_{40}$ *in vitro*. Fibrillation kinetic assays of bicine-treated $A\beta_{40}$ show that aggregation occurs in a time- and concentration-dependent manner with high reproducibility. Sufficient β-sheet formation during an incubation period of 2 days in 20 mM of bicine was confirmed via CD spectroscopy and TEM imaging. Furthermore, prolonged incubation in bicine was found to increase fibrillar forms of $A\beta_{40}$ and also reverse the trend of toxicity. By adjusting the incubation period of $A\beta_{40}$ in bicine, the fibril-to-oligomer composition as well as the cytotoxicity of the aggregates can be regulated.

Modifying environmental factors can vastly change the propensity of amyloid aggregation. Slow acquisition of fibrillar $A\beta_{40}$ due to long aggregation periods can be overcome by taking advantage of buffer ions that facilitate fibrillation. Our results show that aggregation conditions with bicine buffer can rapidly acquire β-sheet-rich $A\beta_{40}$ aggregates that possess toxicity. Although we did not observe any side effects by bicine during the acquirement of $A\beta_{40}$ fibrils in this study, there is a need to consider the possibility of unwanted side effects before applicating bicine to different protocols.

## Supporting information

**S1 Fig. Full images of SDS-PAGE gels.** Aβ(1–40) (50 μM) peptides were dissolved and incubated in 6 different buffers, including bicine (20 mM), PBS (1X) and 4 unreported compounds (1, 3, 4, and 6), at 3 time points: 2, 4 and 7 days. The 0 day Aβ(1–40) samples were dissolved in the same buffers and immediately stored at −70˚C before use. Aβ aggregates were visualized by silver staining after SDS-PAGE with PICUP. These experiments were performed in triplicate independently. SDS-PAGE: sodium dodecyl sulfate-polyacrylamide gel electrophoresis, PICUP: photo-induced cross-linking of unmodified proteins, d: day.
(TIF)

**S2 Fig. Concentration-dependent MTT assay.** Assessing bicine-induced cytotoxicity via MTT cell viability assay. Increasing concentrations of bicine was treated to HT-22 cells. Non-treated cells were used as a control. These experiments were performed in triplicate independently. Error bars indicate the standard deviation. Statistical analysis was performed using one-way ANOVA followed by Tukey's multiple comparison test. $^{**}P \leq 0.01$. Conc.: concentration, d: day, Ctrl: control, d: day.
(TIF)

## Acknowledgments

All experimental protocols were approved by Yonsei University. Authors appreciate Chinho Shin from Korea Institute of Science and Technology for CD instrument training.

## Author Contributions

**Conceptualization:** Hye Yun Kim, YoungSoo Kim.

**Funding acquisition:** YoungSoo Kim.

**Investigation:** Hye Yun Kim, HeeYang Lee, Jong Kook Lee, Hyunjin Vincent Kim.

**Methodology:** Hye Yun Kim, HeeYang Lee, Jong Kook Lee, Hyunjin Vincent Kim.

**Supervision:** YoungSoo Kim.

**Visualization:** Hye Yun Kim, HeeYang Lee, Jong Kook Lee, Hyunjin Vincent Kim.

**Writing – original draft:** Hye Yun Kim, HeeYang Lee, Jong Kook Lee, Hyunjin Vincent Kim, Key-Sun Kim, YoungSoo Kim.

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
