## [Decision Letter · Decision Letter 0]

23 Sep 2020

PONE-D-20-20947

Bicine promotes rapid formation of β-sheet-rich amyloid-β fibrils

PLOS ONE

Dear Dr. YoungSoo Kim,

Thank you for submitting your manuscript to PLOS ONE. After careful consideration, we feel that it has merit but does not fully meet PLOS ONE’s publication criteria as it currently stands. Therefore, we invite you to submit a revised version of the manuscript that addresses the points raised during the review process.

Please change your manuscript according to the reviewer's comments.

We look forward to receiving your revised manuscript.

Kind regards,

Eugene A. Permyakov, Ph.D., Dr.Sci.

Academic Editor

PLOS ONE

Journal Requirements:

Reviewers' comments:

Reviewer's Responses to Questions

**Comments to the Author**

1. Is the manuscript technically sound, and do the data support the conclusions?

Reviewer #1: Yes

2. Has the statistical analysis been performed appropriately and rigorously? 

Reviewer #1: N/A

3. Have the authors made all data underlying the findings in their manuscript fully available?

Reviewer #1: Yes

4. Is the manuscript presented in an intelligible fashion and written in standard English?

Reviewer #1: Yes

5. Review Comments to the Author

Reviewer #1: The authors describe a chemical model of Aβ40 oligomer aggregation, the leading hypothesis of the mechanism of cerebral amyloid angiopathy, overlapping in many aspects with Altzheimer’s disease. Their findings allows for observations for chemically controlled changes in the aggregation state of amyloid-beta peptide. This might potentially assist in the development of the drugs targeting the corresponding neuropathologies.

The study, albeit relatively simple, is well planned and conducted. I couldn’t find too many contradicting points in the authors’ logics. Therefore I could recommend the manuscript for publication after a minor revision according to the notes below.

Minor and technical comments

1. Ln. 137, 153 and elsewhere: “…serial concentrations of bicine (0.07813 to 20 mM)…” — the precision of concentrations must be expressed uniformly, to a certain number of significant digits.

2. Ln. 155 and elsewhere — the PBS concentration is not disclosed in the manuscript.

3. Fig. 2. The figure’s title must relate to both buffers used, not to bicine only.

6. PLOS authors have the option to publish the peer review history of their article (what does this mean?). If published, this will include your full peer review and any attached files.

Reviewer #1: No

---

## [Author Response · Author response to Decision Letter 0]

28 Sep 2020

Response to Editor's and Reviewer’s Comments:

Original uncropped and unadjusted images of gels are now provided in Supporting Information. Also, figure 3 was adjusted accordingly to gel reporting requirements of PLOS ONE. As our experiment did not use any human or animal subjects and/or tissue, we did not provide an ethics statement.

Minor Comments

1. Ln. 137, 153 and elsewhere: “…serial concentrations of bicine (0.07813 to 20 mM)…” — the precision of concentrations must be expressed uniformly, to a certain number of significant digits.

- Thank you for your comments. We have accordingly changed the expression of significant figures in a uniform manner.

2. Ln. 155 and elsewhere — the PBS concentration is not disclosed in the manuscript.

- Thank you for your comment. We have added the concentration of PBS (1X) to the materials and methods and figure legend sections in this manuscript.

Ln. 68: “PBS (1X) consists of phosphate buffer concentration of 0.01M and a sodium chloride concentration of 0.154M.” (Added)

3. Fig. 2. The figure’s title must relate to both buffers used, not to bicine only.

- Thank you. We have inserted “PBS” into the title of Fig. 2.

---

## [Editor Report · Decision Letter 1]

30 Sep 2020

Bicine promotes rapid formation of β-sheet-rich amyloid-β fibrils

PONE-D-20-20947R1

Dear Dr. Kim,

We’re pleased to inform you that your manuscript has been judged scientifically suitable for publication and will be formally accepted for publication once it meets all outstanding technical requirements.

Kind regards,

Eugene A. Permyakov, Ph.D., Dr.Sci.

Academic Editor

PLOS ONE
---

## [Editor Report · Acceptance letter]

2 Oct 2020

PONE-D-20-20947R1 

Bicine promotes rapid formation of β-sheet-rich amyloid-β fibrils 

Dear Dr. Kim:

I'm pleased to inform you that your manuscript has been deemed suitable for publication in PLOS ONE. Congratulations! Your manuscript is now with our production department. 

Kind regards, 

on behalf of

Prof. Eugene A. Permyakov 

Academic Editor

PLOS ONE